# Effect of Postoperative Compression Therapy on the Success of Liposuction in Patients with Advanced Lower Limb Lymphedema

**DOI:** 10.3390/jcm10214852

**Published:** 2021-10-22

**Authors:** Shuhei Yoshida, Isao Koshima, Hirofumi Imai, Solji Roh, Toshiro Mese, Toshio Uchiki, Ayano Sasaki, Shogo Nagamatsu

**Affiliations:** 1 The International Center for Lymphedema, Hiroshima University Hospital, 1-2-3, Kasumi, Minami-ku, Hiroshima 734-8551, Japan; koushimaipla@gmail.com (I.K.); imaih61@hiroshima-u.ac.jp (H.I.); solji6004@yahoo.co.jp (S.R.); mese.toshiro1818@me.com (T.M.); 2 Plastic and Reconstructive Surgery, Hiroshima University, Hiroshima 739-8511, Japan; toshio.uchiki@gmail.com (T.U.); vin.pichon.ayano@gmail.com (A.S.); shogonagamatsu@gmail.com (S.N.)

**Keywords:** lymphedema, liposuction, compression therapy, lower limb, postoperative care

## Abstract

Objective: There is limited information on postoperative care after liposuction for lymphedema limb. The aim of this retrospective study was to identify the threshold compression pressure and other factors that lead liposuction for lower limb lymphedema to success. Materials and Methods: Patients were divided according to whether they underwent compression therapy with both stockings and bandaging (SB group), stockings alone (S group), or bandaging alone (B group) for 6 months after liposuction. The postoperative compression pressure and rate of improvement were compared according to the postoperative compression method. We also investigated whether it was possible to decrease the compression pressure after 6 months. Liposuction was considered successful if improvement rate was >15. Results: Mean compression pressure was significantly lower in the S group than in the SB group or B group. The liposuction success rate was significantly higher in the SB group than in the B group or S group. There was not a significant difference between the values at 6 months after liposuction and at 6 months after a decrease in compression pressure in the successful group. Conclusion: Our results suggest that stable high-pressure postoperative compression therapy is key to the success of liposuction for lower limb lymphedema and is best achieved by using both stockings and bandages. The postoperative compression pressure required for liposuction to be successful was >40 mmHg on the lower leg and >20 mmHg on the thigh. These pressures could be decreased after 6 months.

## 1. Introduction

Lymphedema is divided into primary and secondary forms based on the underlying etiology. The vast majority of lymphedema worldwide is secondary lymphedema. In industrialized countries, cancer therapy is the leading cause of secondary lymphedema. Approximately 30% of patients who have undergone breast cancer surgery develop lymphedema of the upper limb [1]. Furthermore, 10–30% of patients with gynecological cancer develop lymphedema [2,3,4]. Recently, the Da Vinci surgical robot has been used for lymph node dissection and preventing the development of lymphedema [5]. The surgical procedures used to treat lymphedema are typically categorized as a physiological reconstruction (using lymphovenous anastomosis (LVA), or a vascularized lymphatic transplant (VLT)), or debulking using liposuction or direct excisional procedures [6]. Surgical procedures are typically used when physiotherapy and compression therapy has been inadequate. LVA is often used when the lymphatics are mildly impaired, and VLT and liposuction when lymphatic impairment is more advanced [7,8]. However, patients with chronic advanced lymphedema, in whom lymphatic stasis and impairment is accompanied by deposition of fibroadipose soft tissue [9,10], require debulking surgery [6,11]. Liposuction is the most common type of debulking surgery performed in these patients, and its impact has been debated since the late 1980s [12,13,14].

Liposuction for lymphedema is followed by a rigorous compression therapy as part of routine postoperative care [14,15,16]. Thus, we speculated that effective compression therapy may be critical to the success of liposuction for lymphedema. However, there are limited relevant data on postoperative care to optimize the chances of liposuction being successful in patients with lower limb lymphedema, in particular the amount of compression pressure required and how it is best achieved. In this study, we retrospectively reviewed our patients with lower limb lymphedema who had been treated by liposuction and sought to identify factors that lead liposuction to success.

## 2. Methods and Patients

This retrospective study included patients with lower limb lymphedema treated by liposuction at Hiroshima University Hospital between May 2019 and May 2020. All patients had been treated by LVA and VLT before liposuction but without an adequate volume reduction. The study was approved by our institutional review board (approval number: E-1413) and conducted in accordance with the Declaration of Helsinki and the STROBE guidelines (http://www.strobe-statement.org/, accessed on 7 April 2019). All study participants provided written informed consent. 

Preoperatively, the stocking compression pressure was adjusted to between 20 mmHg and 30 mmHg on the lower leg and between 10 mmHg and 15 mmHg on the thigh at the highest pressure the patient could tolerate (JOBST^®^, BSN-JOBST GmbH, Hamburg, Germany). 

ICG lymphography images were recorded in the plateau phase (i.e., 12–18 h after the injection or on the following day).

Lymphoscintigraphy was also performed to assess lymphatic function in the affected limb and obtained images of the limb at 10, 60, and 120 min after injection. 

The inclusion criteria were a diffuse lymphedema pattern throughout the lower limb, seen on lymphography; type III lymphedema on lymphoscintigraphy, using the classification described by Maegawa et al. [17]; ISL (International Society of Lymphology) late stage II–III lymphedema [18]; and no weight gain during the year before and after liposuction.

The volume of the affected lower limb was evaluated by measuring the circumference at 5 anatomic locations (10 cm above the knee, at the knee, and 10 cm below the knee, ankle, and foot) before and 1 year after surgery in the supine position after confirming that there was no cellulitis. The lower extremity lymphedema (LEL) index was calculated by dividing the sum of the squares of the circumference in the 5 areas of the affected extremity by the BMI [19]. The rate of improvement in lymphedema was calculated by dividing the difference in the LEL index between, before and after surgery by the preoperative value for each case as follows: ([preoperative LEL index] − [postoperative LEL index])/(preoperative LEL index) × 100. 

The compression pressure was measured at the front midpoint on the lower leg and on the thigh using a portable pressure sensor (Palm Q^®^, Cape Co. Ltd., Yokosuka, Japan).

### 2.1. Surgical Procedures

#### Physiological Reconstruction

LVA and VLT were performed at least once in all cases before liposuction. 

In the absence of a defined pattern, incisions were made along the anatomic courses of the superficial veins, preferably along the greater saphenous vein. Whenever healthy lymphatic vessels and size-matched veins were present, LVAs were performed. 

All vascularized lymphatic tissue transfers were performed by perforator–to–perforator anastomosis under general anesthesia and as close as possible to the joint that was most troublesome for the patient, using various types of adipo-lymphatic perforator tissue flaps that were harvested, including a superficial circumflex iliac artery perforator flap [20], a first metatarsal artery flap [21], and a lateral thoracic artery perforator flap [22]. A subcutaneous vein was used for the venous anastomoses. The total number of LVAs or VLTs in each lower limb before liposuction were recorded.

### 2.2. Liposuction

After LVA or VLTs, extensive lipectomy was performed while sparing the LVA or VLT sites in order to avoid injury [23,24,25]. 

A tourniquet was used in combination with the tumescent technique (subcutaneous infusion of 1000 mL of saline mixed with 1 mg of adrenaline and 40 mL of lidocaine 1% (Xylocaine, Aspen Japan K.K., Tokyo, Japan) to minimize blood loss [26,27]. Approximately 10 incisions, each 3–4 mm long, were made and liposuction was performed using cannulas that were 15 cm or 25 cm in length with a diameter of 6 mm. Circumferential liposuction was performed from the ankle to the hip, and as much of the hypertrophied fat was removed as possible using the previously measured circumferences in the healthy limb as reference values. When the areas distal to the tourniquet had been treated, a cotton bandage was applied to the limb to minimize bleeding and reduce postoperative edema. The tourniquet was then removed, and the most proximal portion of the limb was treated using the tumescent technique. Finally, the proximal part of the compression bandage was rolled up to compress the proximal portion of the limb and secured in position on the trunk using adhesive tape. When the proximal portion was too large for the tourniquet to compress the limb sufficiently to stop bleeding [28], liposuction was performed first, using the tumescent technique to reduce the volume of the proximal portion. The distal portion of the limb was then treated using a tourniquet. The incisions were left open for drainage of subcutaneous hemorrhage after surgery. The volume of the fat removed was calculated. 

### 2.3. Postoperative Management

Compression was applied for 7 days after liposuction using a cotton bandage that was not changed during that time. The pressure was adjusted to be ≥40 mmHg on the lower leg and ≥20 mmHg on the thigh.

The patients were divided into three groups according to the type of postoperative compression therapy applied after hospitalization: a compression stocking (JOBST Bellavar^®^) and bandaging (JOBST Comprihaft^®^) with the pressure adjusted to ≥40 mmHg on the lower leg and ≥20 mmHg on the thigh (SB group, *n* = 11); a compression stocking only with the pressure adjusted to 20–30 mmHg on the lower leg and 10–15 mmHg on the thigh (S group, *n* = 5); and thin cotton bandaging only with the pressure adjusted to ≥40 mmHg on the lower leg and ≥20 mmHg on the thigh (B group, *n* = 5). All patients were instructed that the compression garments should be worn continuously both night and day. 

### 2.4. Follow-Up 

Patients were followed up at 0.5, 1, 1.5, 2, 3, 6, 7, 8, 9, and 12 months after liposuction surgery. At each visit, their compliance with application of compression pressure, garment slipping, pain, and skin ulceration were checked by interview and physical examination. The compression pressure applied was also measured at each visit, and the average of the measurements recorded between 0.5 and 6 months was calculated for each patient. After 6 months of compression, the pressure applied was decreased to 20 mmHg on the lower leg and 10 mmHg on the thigh using a compression stocking (JOBST Bellavar^®^).

### 2.5. Evaluation of Volume Reduction

Images of the lower limbs obtained before and a year after liposuction were compared by three plastic surgeons working independently. Liposuction was deemed to be successful or unsuccessful based on whether all three observers judged that there was a volume reduction in the affected limb a year later. The rate of improvement in the LEL index was then compared between the successful and unsuccessful groups.

### 2.6. Statistical Analysis

The data are shown as the mean and standard deviation (range). The Tukey–Kramer test was used to compare the mean compression pressure measured during the year after liposuction, the rate of improvement in the LEL index, and the total liposuction volume according to the postoperative compression method used. The liposuction success rate was compared according to the type of postoperative management using the chi-squared test. Compression pressure, total liposuction volume, the total number of LVAs or VLTs in each lower limb, and BMI were compared according to whether liposuction was successful or unsuccessful using the Student‘s *t*-test. Changes in the LEL index at 6 months after liposuction (immediately before a decrease in the compression pressure), and at 1 year after liposuction (6 months after the decrease in the compression pressure) were compared in the whole limb, the successful group and the unsuccessful group using repeated-measures single-factor analysis of variance. All statistical analyses were performed using Statcel 4 software (OMS Publishing, Inc., Tokyo, Japan). A *p*-value of <0.05 was considered statistically significant.

## 3. Results

Nineteen patients with lymphedema affecting 21 lower limbs were enrolled in the study (Table 1). All surgical procedures were performed without postoperative complications, including lymphorrhea, except for one patient who developed a 5 × 3-cm skin ulcer on the lateral aspect of the lower limb after liposuction that was treated conservatively and took 2 months to epithelialize. There were no cases of cellulitis during the study period.

Volume reduction was judged to be successful when the rate of improvement in the LEL index was >15; therefore, this value was considered to be the threshold for determining whether liposuction was successful (Figure 1).

LEL; lower extremity lymphedema, LVA; lymphovenous anastomosis, VLT; vascularized lymphatic transplant.

The mean compression pressure applied to the lower leg and thigh in the 6 months after liposuction was significantly lower in the S group than in the SB group (*p* < 0.01) and the B group (*p* < 0.01) (Figure 2 and Figure 3). There was no significant difference among the three groups in the mean compression pressure applied to the lower leg (*p* = 0.37) or thigh (*p* = 0.35) after the compression pressure was decreased (Figure 4 and Figure 5). 

The rate of improvement in the LEL index was significantly lower in the S group than in the SB group and in the B group (both *p* < 0.001) with no significant difference between the SB and B groups (Figure 6). However, the liposuction success rate was significantly higher in the SB group than in S group, and there was a difference among the three groups (Figure 7).

### Type of Compression Used in Unsuccessful Cases 

The only unsuccessful case in the SB group was the patient who developed the skin ulcer. There were three unsuccessful cases in the B group. Two of these patients reported frequently slipping off the bandage during the first 2 postoperative months. The third unsuccessful case in the B group was a patient who persisted in wrapping an excessive amount of thin cotton cushioning around the lower limb under the bandage (Table 2).

The average compression pressure measured during the initial 6 months after liposuction was significantly higher in the successful group than in the unsuccessful group (thigh, *p* = 0.001; lower leg, *p* = 0.0004; Table 3). 

There was no significant difference in the total liposuction volume among the three groups (Figure 8) or according to whether liposuction was successful or unsuccessful (Figure 9). However, there was a significant difference in BMI between the successful and unsuccessful groups (*p* = 0.03; Figure 10). The total number of LVAs or VLTs in each lower limbs were larger in the successful group than in the unsuccessful group; however, there was no significant difference between the two groups (Figure 11 and Figure 12). 

There was not a significant difference between the value at 6 months after liposuction and that at 6 months after the decrease in compression pressure for the entire lower limb (*p* = 0.41; Figure 13). In the successful group, there was not a significant difference between the values at 6 months after liposuction and at 6 months after a decrease in compression pressure (*p* = 0.37; Figure 14). In the unsuccessful group, these values were not significantly different among the three groups (*p* = 0.07; Figure 15).

## 4. Discussion

In this study, successful volume reduction could not be achieved in patients with lower limb lymphedema who wore a compression stocking only in the 6 months after liposuction. This finding indicates that a greater postoperative compression pressure on the lower leg and thigh is necessary to maintain the effect of volume reduction achieved by liposuction.

The rate of improvement in the LEL index was significantly higher in our SB and B groups than in our S group, with no significant difference between the SB and B groups. However, there was a significant difference in the success rate between the SB and B groups, which indicates that the type of compression applied after surgery influences the success rate of liposuction for lymphedema, even when the same level of compression pressure is applied. The patients in our B group often complained about the stability of their bandage, particularly in the thigh region, despite using an adhesive product. The difference in compression therapy between the SB and B groups was the stability of the method used. The outcome of liposuction was not successful if the bandage slipped off frequently during the daytime. 

These findings suggest that a stable high postoperative compression pressure is key to a successful outcome after liposuction. 

In general, multilayer bandaging is applied in cases of advanced lower limb lymphedema in order to achieve a higher compression pressure [29]. However, a compression bandage loosens easily and needs frequent rewinding [30], which may attenuate the effect of liposuction in the postoperative period. A stocking with high compression pressure is superior to a compression bandage in terms of stability but may be too burdensome for some patients to wear after liposuction.

In order to balance the needs for high pressure, stability, and ease to wear, both a compression stocking and a compression bandage were worn by most patients in this study because of their cumulative effect on compression pressure [31,32,33]. A variation of this method is to wear two compression stockings on the same lower limb, and some reports suggest that a double stocking is superior to stocking and bandaging in terms of stability [31,32,33]. However, most of our patients wore a stocking and bandage after liposuction in order to manage deformity of the lower limb due to lymphedema.

According to our study findings, the threshold postoperative compression pressure is likely to be around 40 mmHg for the lower leg and about 20 mmHg for the thigh for liposuction to be successful.

Although further observation is necessary, our experience suggests that it is possible to decrease the compression pressure over time and that some mechanisms may be involved. First, LVA and VLT were performed at least once before liposuction in all our patients. In our study, there were no cases in which only liposuction was performed. Therefore, it is unclear whether it is possible to decrease the compression pressure after a period of time in patients who have undergone liposuction without LVA or VLT. Moreover, the possibility of decreasing the compression pressure applied after liposuction have not been discussed in the literature, as far as we are aware. In our study, although there was no significant difference between the two groups, the total number of LVAs or VLTs in each lower limb were larger in the successful group than in the unsuccessful group, in particular, the *p*-value was 0.06 in comparison with VLTs. There might be a possibility that an increased number of LVAs and VLTs contribute to a successful result of liposuction.

The second potential mechanism is lymphangiogenesis. Vascular endothelial growth factor (VEGF)-C is well known to promote lymphangiogenesis via VEGFR3 [34,35,36,37] and has been found to be overexpressed in limbs with lymphedema [38,39]. It is not clear whether lymphatics can be damaged during liposuction [40,41]. However, overexpression of VEGF-C in limbs with lymphedema could promote lymphangiogenesis, even after liposuction, and allow an eventual decrease in compression pressure. On the other hand, overexpression of VEGF-C may aggravate lymphedema and promote vascular leakage [38,39]. These conflicting features of VEGF-C may explain the need for a high compression pressure during the initial management after liposuction and the possibility of decreasing the pressure after a certain time. It is still unclear whether peripheral lymphovenous communications (LVC) exist, and if so, how they function [42]. However, experimental and It isclinical studies suggest that obstruction of major lymph channels, which increases intralymphatic volume and pressure, contributes to the functioning of LVC [43,44,45,46,47,48]. Therefore, there may be some relationship between lymphangiogenesis, LVC, and postoperative management after liposuction.

The third mechanism may be related to obesity, which has recently been identified as a risk factor for lymphedema [49,50]. In our study, patients with a high BMI tended to have a poorer postoperative outcome, which suggests that obesity is a risk factor for not only lymphedema itself but also failure of liposuction. The mechanisms via which obesity impairs lymphatic function and the outcome of liposuction are not known. However, it is suspected that the amount of lymph produced by the body increases as BMI increases, and that the greater amount of lymph may interfere with the ambulation/muscle contraction in the lower limbs required to transport the fluid [51].

According to Laplace’s law, a higher pressure occurs inside the lower limb when its circumference is narrower and when the same pressure is applied to the surface of the limb [52,53]. Surface pressures do not accurately reflect underlying soft tissue pressures in that the soft tissue pressure is consistently lower than the surface pressure. The percentage of surface pressure reflected in the underlying tissue varies inversely with the circumference of the limb. Furthermore, there is a tendency for the pressure in soft tissue to decrease with increasing depth from the surface. This tendency is minimal in limbs with a smaller circumference but becomes more pronounced as the circumference of the limb increases [28]. Accordingly, limbs with less soft tissue are more amenable to compression therapy and may explain the effect of liposuction on limbs with lymphedema and enable a decrease in compression pressure after a certain period of time following liposuction. 

The main side-effect is “heat” caused by the stable high-pressure postoperative compression therapy. It is tolerable in cool seasons; however, it is intolerable in summer. Therefore, we recommend performing the liposuction followed by stable high-pressure postoperative compression therapy at the biggening of the cool season. 

In this study, the evaluation of excess fat was performed only by circumferential measurement. However, magnetic resonance imaging, ultrasound, or computed tomography are recommended for accurate evaluation of excess fat in lymphedema in order to distinguish volume change between muscle and fat.

The indication of liposuction for lymphedema treatment is recommended only after physiological reconstruction, using LVA and VLT, are performed several times, because there is a possibility that physiological reconstruction using LVA and VLT are helpful for successful liposuction, even if those procedures did not show direct effect of volume reduction. 

## 5. Conclusions

Our results suggest that stable high-pressure postoperative compression therapy is key to a successful outcome after liposuction in patients with lymphedema. Wearing of both a compression stocking and a bandage is advisable to achieve a stable high pressure. The threshold postoperative compression pressure should be ≥40 mmHg on the lower leg and ≥20 mmHg on the thigh for at least 6 months after surgery.

## Figures and Tables

**Figure 1 jcm-10-04852-f001:**
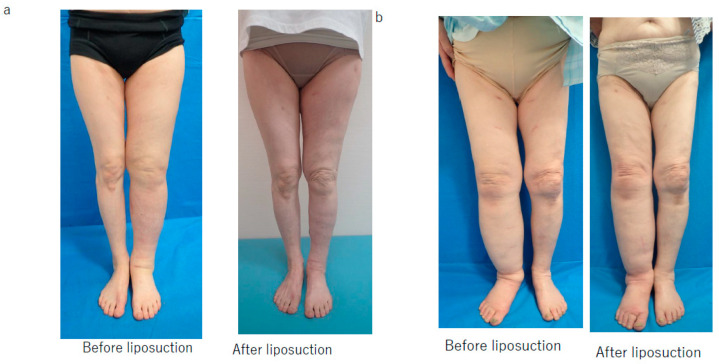
Representative examples of successful and unsuccessful outcomes after liposuction according to type of compression strategy. (**a**) The patient was a 71-year-old woman with secondary lymphedema in the left lower limb in whom there was insufficient improvement after lymphaticovenous anastomosis surgery. After removal of 2000 mL of fat by liposuction and wearing both a compression stocking and bandage for 6 months, her lower extremity lymphedema index improved by 20.4, which was considered a successful outcome. (**b**) The patient was a 78-year-old woman with secondary lymphedema in the right lower limb that did not improve adequately after lymphaticovenous anastomosis surgery. After removal of 1800 mL of fat by liposuction and wearing a bandage alone for 6 months, her lower extremity lymphedema index improved by only 12.3, which was considered an unsuccessful result.

**Figure 2 jcm-10-04852-f002:**
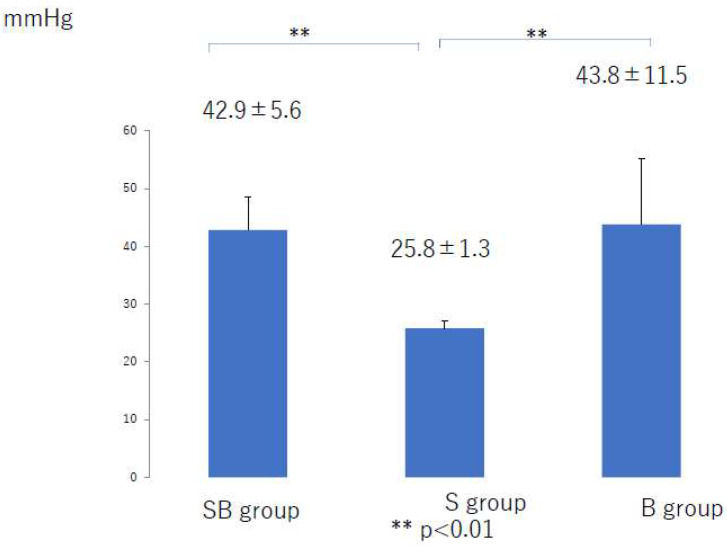
Mean compression pressure applied to the lower limb during the first 6 months after liposuction. B, bandage only; SB, compression stocking and bandage; S, stocking only.

**Figure 3 jcm-10-04852-f003:**
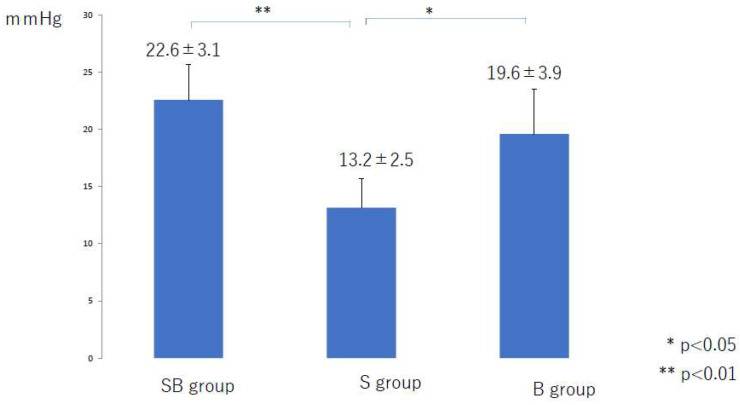
Mean compression pressure applied to the thigh during the first 6 months after liposuction. B, bandage only; SB, compression stocking and bandage; S, stocking only.

**Figure 4 jcm-10-04852-f004:**
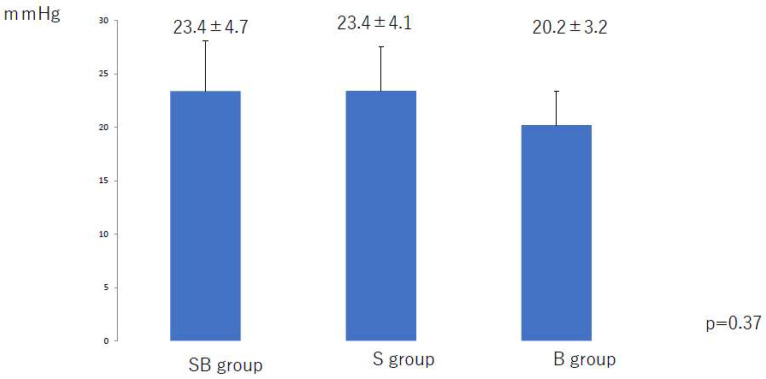
Mean compression applied to the lower leg after a decrease in pressure at 6 months after liposuction. B, bandage only; SB, compression stocking and bandage; S, stocking only.

**Figure 5 jcm-10-04852-f005:**
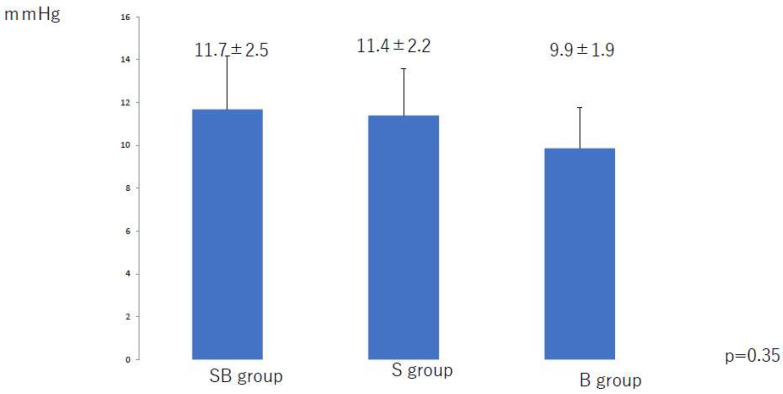
Mean compression applied to the thigh after a decrease in pressure at 6 months after liposuction. B, bandage only; SB, compression stocking and bandage; S, stocking only.

**Figure 6 jcm-10-04852-f006:**
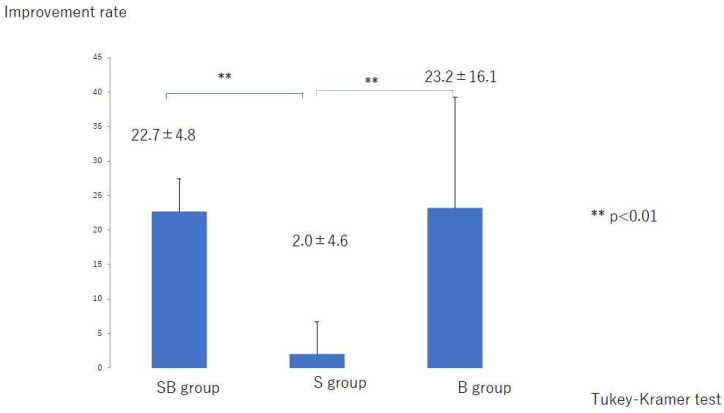
Rate of improvement in the lower extremity lymphedema index. B, bandage only; SB, compression stocking and bandage; S, stocking only.

**Figure 7 jcm-10-04852-f007:**
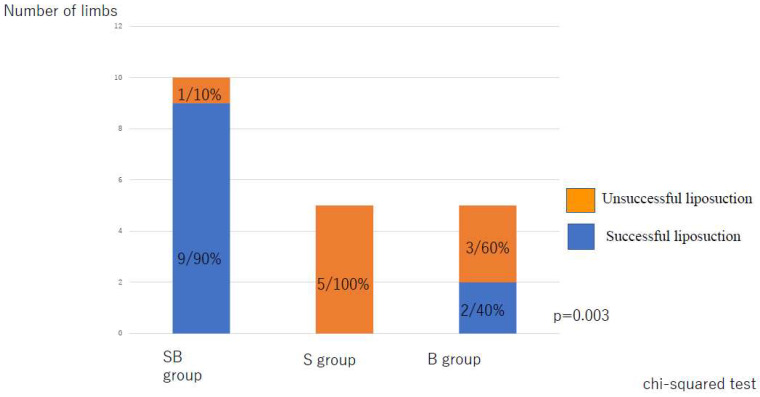
Liposuction success rate according to type of postoperative compression strategy. B, bandage only; SB, compression stocking and bandage; S, stocking only.

**Figure 8 jcm-10-04852-f008:**
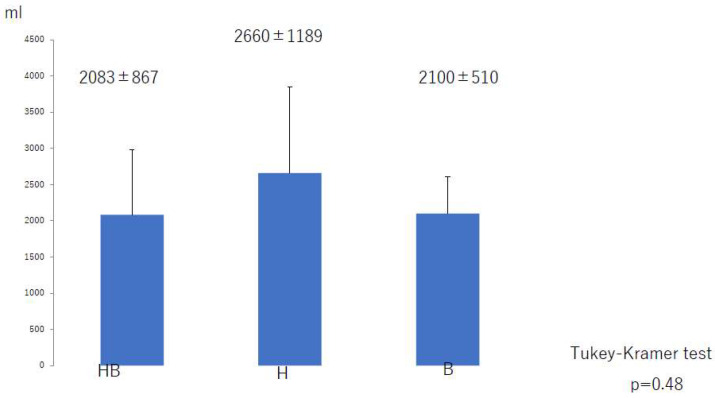
Total liposuction volume according to type of postoperative compression strategy. B, bandage only; SB, compression stocking and bandage; S, stocking only.

**Figure 9 jcm-10-04852-f009:**
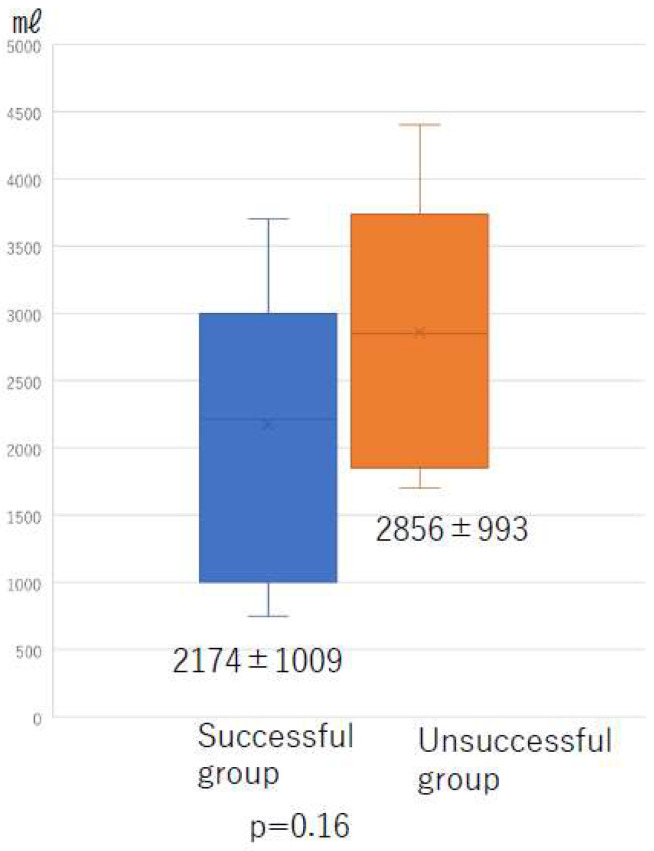
Total amount of fat removed according to whether liposuction was successful or unsuccessful.

**Figure 10 jcm-10-04852-f010:**
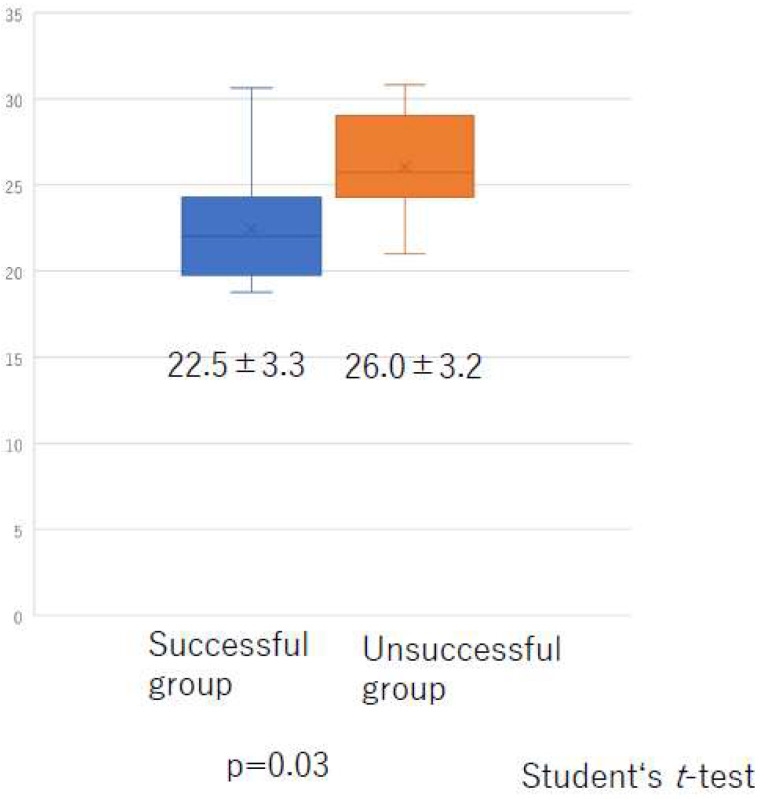
Body mass index according to whether liposuction was successful or unsuccessful.

**Figure 11 jcm-10-04852-f011:**
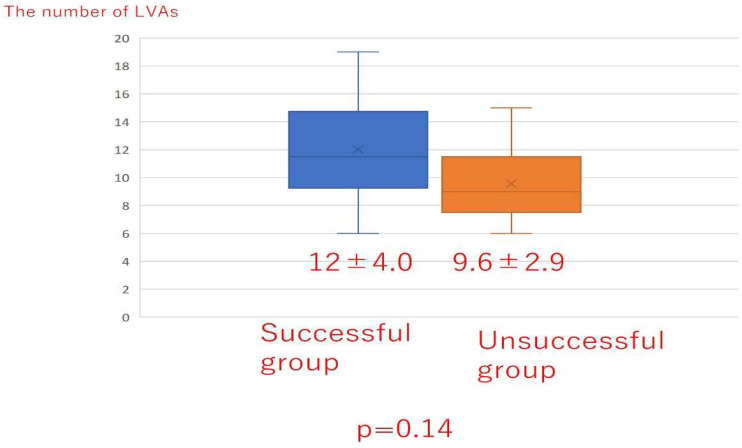
The total number of LVAs in each lower limb according to whether liposuction was successful or unsuccessful. LVA; lymphovenous anastomosis.

**Figure 12 jcm-10-04852-f012:**
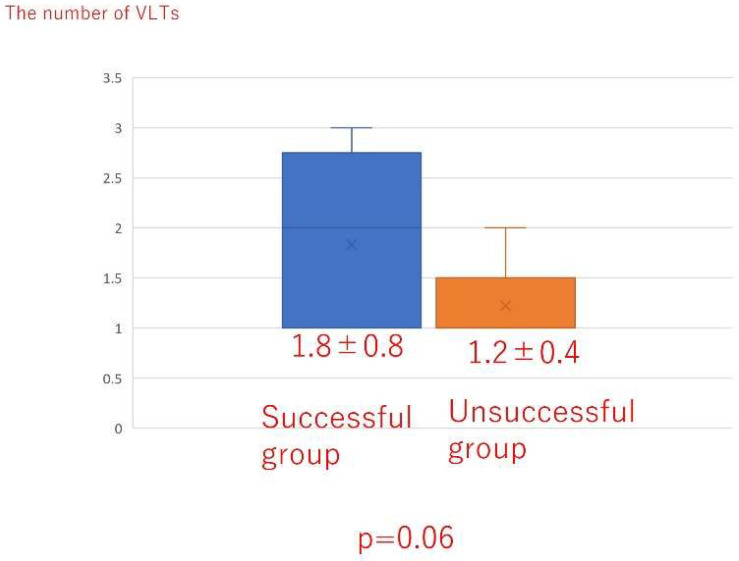
The total number of VLTs in each lower limb according to whether liposuction was successful or unsuccessful. VLT; vascularized lymphatic transplant.

**Figure 13 jcm-10-04852-f013:**
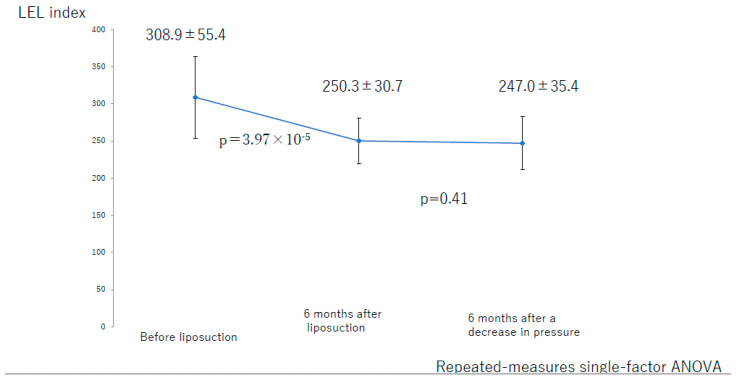
Changes in the LEL index over time in the entire study population. ANOVA, analysis of variance; LEL, lower extremity lymphedema.

**Figure 14 jcm-10-04852-f014:**
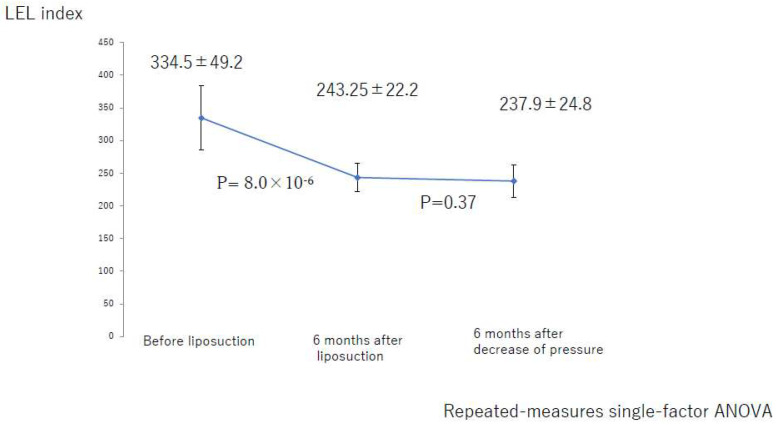
Changes in the LEL index over time in the group with successful liposuction. ANOVA, analysis of variance; LEL, lower extremity lymphedema.

**Figure 15 jcm-10-04852-f015:**
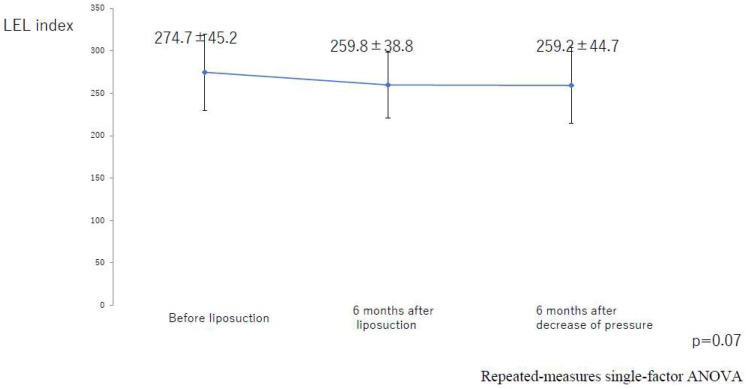
Changes in the LEL index over time in the group with successful liposuction. ANOVA, analysis of variance; LEL, lower extremity lymphedema.

**Table 1 jcm-10-04852-t001:** LEL; Lower Extremity Lymphedema.

Variable	Lower Extremity
Female:male (*n*)	18:1
Primary:secondary lymphedema (*n*)	6:13
Age (years)	59.6 ± 12.5 (35–78)
Body mass index	23.0 ± 3.6 (18.8–30.8)
The number of LVAs in each limbs	11.0 ± 3.7 (6–19)
The number of VLTs in each limbs	1.6 ± 0.7 (1–3)
LEL index before surgery	308.9 ± 55.4 (211–432)
LEL index after surgery	247.0 ± 35.4 (203–314)
Improvement rate (%)	17.9 ± 12.2 (−0.1–44.2)
Total liposuction volume (mL)	2461.1 ± 1034.0 (750–4400)
Follow-up duration, (months)	15.3

**Table 2 jcm-10-04852-t002:** Type of compression strategy used in patients in whom liposuction was unsuccessful.

Strategy	Pressure (mmHg)		Cause
	Lower Leg	Thigh	
Stocking + bandage	31.7 ± 6.1 (22–38)	16.3 ± 2.9 (12–20)	Skin ulcer
Bandage only	43.8 ± 3.9 (40–50)	21.0 ± 2.4 (18–25)	Bandage slipping off
Bandage only	41.5 ± 2.7 (39–46)	20.3 ± 2.1 (18–23)	Bandage slipping off
Bandage only	21.8 ± 3.1 (19–26)	10.3 ± 2.1 (8–13)	Excessive cotton cushioning applied

**Table 3 jcm-10-04852-t003:** Compression pressures applied according to whether liposuction was successful or not.

	Successful	Unsuccessful	*p*-Value
Thigh (mmHg)	22.4 ± 3.0	14.5 ± 6.1	0.001
Lower leg (mmHg)	44.4 ± 5.4	26.5 ± 12.7	0.0004

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
