# Peer review of "Effect of Postoperative Compression Therapy on the Success of Liposuction in Patients with Advanced Lower Limb Lymphedema"

_jcm, 2021, doi:10.3390/jcm10214852_

Round 1

Reviewer 1 Report

The study clearly shows the importance of compression mechanisms in the post-operative period of these patients. Higher compression pressures have superior results. Pressures greater than 40 mmHg are suggested, but above the knee it is difficult to achieve and maintain these pressures.
Patient adherence, at higher pressures 40, in daily clinical practice, is lower and this is another aspect to be considered and analyzed in the surgical indications of these patients. Studies show these aspects and normally a minimum pressure to maintain the results in the treatment of lymphedema is around 30 mmHg and that it is difficult to achieve in the thigh where the treatment failure can be greater.

Author Response

Reviewer 1:

The study clearly shows the importance of compression mechanisms in the post-operative period of these patients. Higher compression pressures have superior results. Pressures greater than 40 mmHg are suggested, but above the knee it is difficult to achieve and maintain these pressures.

Patient adherence, at higher pressures 40, in daily clinical practice, is lower and this is another aspect to be considered and analyzed in the surgical indications of these patients. Studies show these aspects and normally a minimum pressure to maintain the results in the treatment of lymphedema is around 30 mmHg and that it is difficult to achieve in the thigh where the treatment failure can be greater.

I am sorry, but there may be some misunderstanding. Pressure greater than 40mmHg is applied to only below knee. Pressure above knee is lower than 25mmHg.

Reviewer 2 Report

The authors retrospectively reviewed patients with lower limb lymphedema who had been treated by liposuction and sought to identify factors that lead liposuction to success.

They suggest that stable high-pressure postoperative compression therapy is key to a successful outcome after liposuction in patients with lymphedema.

The authors need to discuss the following points additionally, in order to increase the persuasiveness of this paper

  1. In this study, the authors used the results of ICG lymphography and lymphoscintigraphy for  inclusion criteria. Usually, MRI or ultrasound are often used for evaluate of excess fat in lymphedema. ICG lymphography and lymphoscintigraphy can evaluate the function of lymphatic vessels and lymph leaks not excess fat. Why did not the authors use MRI or ultrasound for inclusion criteria and evaluation of post operative results ?
  2. The authors mentioned that LVA and VLT were performed at least once in all cases before liposuction. The authors should clarify the relationship between the results of this study and pre-performed LVA and VLT with some statistics because there is possibility that the number of LVAs and VLTs affects the results of liposuction.  

Author Response

Thank you for review. The attached Word file is author's notes to Reviewer.

Reviewer 3 Report

Dear Authors:

In the manuscript by Yoshida et al. the authors suggest that stable high-pressure postoperative compression therapy is key to a successful outcome after liposuction in patients with lymphedema. I would give just a few suggestions.

1. Some citations are missing: 

In page 11, line 318-319:"The surgical procedures used to treat lymphedema are typically categorized as physiological reconstruction (using lymphovenous anastomosis [LVA] or vascularized lymphatic transplant [VLT]) or debulking using liposuction or direct excisional procedures(1). Surgical procedures are typically used when physiotherapy and compression therapy has been inadequate."  Recently the Da Vinci surgical robot has been used for lymph node dissection and preventing the development of lymphedema. (please cite, Chen K et al. Efficacy of da Vinci robot-assisted lymph node surgery than conventional axillary lymph node dissection in breast cancer - A comparative study. Int J Med Robot. 2021 Jul 16:e2307. doi: 10.1002/rcs.2307.) 

2. In this study, whether the author has observed or investigated the potential side-effects of stable high-pressure postoperative compression therapy, which may reduce patient compliance? If yes, please clarify or discuss about it.

Best,

Author Response

Thank you for review. The attached file is author`s notes to reviewer.

Round 2

Reviewer 2 Report

Thank you for quick response for review.

Please mention about evaluation of excess fat and surgical indication of  liposuction in discussion for not misleading the readers. 

Reviewer 3 Report

Authors made correction according to my previous suggestions. Strongly recommend for publishing.

Sincerely,

Author Response

Thank you very much. We are happy!